# Density Gradient Centrifugation Is an Effective Tool to Isolate Cancer Stem-like Cells from Hypoxic and Normoxia Triple-Negative Breast Cancer Models

**DOI:** 10.3390/ijms25168958

**Published:** 2024-08-17

**Authors:** Camillo Sargiacomo, Aleksandr Klepinin

**Affiliations:** 1Translational Medicine, School of Science, Engineering and Environment (SEE), University of Salford, Greater Manchester, Salford M5 4WT, UK; c.sargiacomo@salford.ac.uk; 2Laboratory of Chemical Biology, National Institute of Chemical Physics and Biophysics, 12618 Tallinn, Estonia

**Keywords:** triple-negative breast cancer, oxidative phosphorylation, glycolysis, cancer stem-like cells (CSCs), metabolic plasticity, hypoxia

## Abstract

Accumulating evidence has indicated that stemness-related genes are associated with the aggressiveness of triple-negative breast cancer (TNBC). Because no universal markers for breast CSCs are available, we applied the density gradient centrifugation method to enrich breast CSCs. We demonstrated that the density centrifugation method allows for the isolation of cancer stem cells (CSCs) from adherent and non-adherent MCF7 (Luminal A), MDA-MB-231 (TNBC) and MDA-MB-468 (TNBC) breast cancer cells. The current study shows that the CSCs’ enriched fraction from Luminal A and TNBC cells have an increased capacity to grow anchorage-independently. CSCs from adherent TNBC are mainly characterized by metabolic plasticity, whereas CSCs from Luminal A have an increased mitochondrial capacity. Moreover, we found that non-adherent growth CSCs isolated from large mammospheres have a higher ability to grow anchorage-independently compared to CSCs isolated from small mammospheres. In CSCs, a metabolic shift towards glycolysis was observed due to the hypoxic environment of the large mammosphere. Using a bioinformatic analysis, we indicate that hypoxia HYOU1 gene overexpression is associated with the aggressiveness, metastasis and poor prognosis of TNBC. An in vitro study demonstrated that HYOU1 overexpression increases breast cancer cells’ stemness and hyperactivates their metabolic activity. In conclusion, we show that density gradient centrifugation is a non-marker-based approach to isolate metabolically flexible (normoxia) CSCs and glycolytic (hypoxic) CSCs from aggressive TNBC.

## 1. Introduction

Breast cancer is the most diagnosed tumor and the main cause of death among females worldwide [1]. Traditional therapy for breast cancer usually targets hormone receptors (the estrogen receptor (ER) and progesterone receptor (PR)) or epidermal growth factor receptor 2 (HER2). However, common chemotherapy is inefficient against this breast subtype, which lacks ER, PR and HER2 expression, known as triple-negative breast cancer (TNBC). To make the situation worse, TNBC is the most aggressive breast cancer subtype [2]. Therefore, the development of a new therapeutic strategy is crucial for TNBC. 

One of the key cancer hallmarks is the ability of malignant cells to rearrange their energy metabolism to satisfy the demands of proliferation and metastatic activity [3,4]. A large body of evidence shows the ability of cancer cells to navigate their metabolism between glycolysis and oxidative phosphorylation (OXPHOS), switching from one state to another in order to adapt to the fluctuating tumor microenvironment and survive [5,6,7]. Based on an in vitro study, the hybrid glycolysis/OXPHOS metabolic model was proposed for TNBC [8]. The metabolic plasticity (hybrid metabolic state) of cancer cells that allows for a shift towards glycolysis is one of the main reasons why malignant breast cells can survive under intermitted hypoxic conditions [9]. Moreover, a hypoxic microenvironment may help to enrich the tumor with poorly differentiated/undifferentiated cancer cells and prevent cellular differentiation. As a result, a hypoxic niche accelerates the formation of a small sub-population of cancer stem cells (CSCs), which are the main root cause of tumor recurrence and metastasis [10]. A recent study demonstrated that stemness-related genes are associated with the aggressiveness of TNBC and that stemness markers have prognostic value for TNBC [11]. Our study on a Luminal A breast cancer model showed that chronic hypoxia increases the stemness of cancer cells [12]. 

In general, there are two types of hypoxias: acute and chronic. Acute hypoxia usually occurs at the early stage of cancer due to rapid malignant cell proliferation, whereas chronic hypoxia develops at the late stage of cancer due to aberrant tumor vasculature [13]. In vitro studies have shown that the regulation of malignant progression, invasion and metabolic rearrangement mainly involves hypoxia-inducible factors (HIFs) and the hypoxia up-regulated 1 (*HYOU1*) gene [13,14]. Under hypoxic conditions, HYOU1, also known as Orp150, serves as a molecular chaperon in the endoplasmic reticulum [14]. There is evidence showing that *HYOU1* is overexpressed in several cancers, including breast cancer [14]. Our study on a Luminal A breast cancer model demonstrated that chronic hypoxia increases the stemness of cancer cells via the overexpression of mitochondria-related genes [12]. A recent work on lung multicellular tumor spheroid revealed that the downregulation of *HYOU1* suppresses the stemness and malignancy of lung cancer [15]. However, it is still unclear whether the alteration in *HYOU1* affects the stemness and metabolism of TNBC cells.

Isolated breast CSCs are good in vitro models for developing a new therapeutic strategy to eradicate the root cause of cancer [16]. Different separation techniques are available to isolate breast CSCs. Cell-surface phenotype separation is one of the common methods to isolate CSCs. Cell-surface markers like CD44+/CD24− and cytosolic stemness marker ALDH+ are widely used for the enrichment of CSCs from breast cancer [17]. However, there are no universal CSC markers available for different breast cancer subtypes [18]. An alternative simple method to enrich breast CSCs is needed. According to the CSCs theory, cell asymmetric division increases intra-tumoral heterogeneity and helps to form a sub-population of CSCs [19]. Therefore, a method for the isolation of CSCs is based on their physical properties. For instance, several works demonstrated that the density gradient centrifugation method is an efficient tool to isolate CSCs from a primary hepatic cancer rat model [20] and human primary glioblastoma tumor [21]. 

In the present study, we aimed to determine whether breast CSCs can be physically separated from adherent and non-adherent breast cancer models. For this purpose, we applied the density gradient centrifugation method using Luminal A and TNBC models. After centrifugation, four to six fractions were collected. The second fraction contained of CSCs enriched subpopulation, showing an increased capacity to grow anchorage-independently. We demonstrated that CSCs isolated from the adherently grown Luminal A had increased mitochondrial capacity, whereas CSCs isolated from adherently grown TNBC showed metabolic plasticity. In contrast, CSCs isolated from large mammospheres shifted their metabolism towards glycolysis most probably due to the hypoxic environment inside of mammospheres. We demonstrated that the tumor hypoxic environment increases breast cancer cells’ stemness via the overexpression of *HYOU1* and hyperactivates their metabolic activity, leading to the aggressiveness of TNBC. In conclusion, density gradient centrifugation represents a non-marker-based approach to isolate metabolically flexible (normoxic) CSCs and glycolytic (hypoxic) CSCs, which are important targets for future therapies for aggressive TNBC.

## 2. Results

### 2.1. Fractionation of Breast Cancer Cells by Density Gradient Centrifugation and CSCs’ Enriched Fraction Determination via Non-Adherent 3D Mammosphere Assay

Numerous studies have shown that the density gradient centrifugation method can be applied for the isolation of CSCs and senescence cells from cancer [20,21,22]. In the current study, the density gradient centrifugation protocol [22] was adapted for breast cancer cells. Here, we used three breast cancer cell lines which represent two subtypes of breast cancer: MCF 7, representing Luminal A, or MDA-MB-231 and MDA-MB-468, representing TNBC. After density gradient centrifugation, six fractions (FI-VI) were gently collected from the top to the bottom based on the density gradient (Figure 1A). Then, the cell distribution within the density gradient was analyzed. Cell counting revealed that the cells from all three cell lines were distributed similarly. Cell segregation from F1 to FVI was as follows: 2.21–3.86% (FI), 5.03–8.32% (FII), 51.98–67.91% (FIII), 13.21–24.54% (FIV), 3.03–4.12% (FV) and 8.19–9.34% (FVI) (Figure 1A). We found that FIII contained the most cells, and FI and FV contained the fewest cells (less than 4%). As the CSC content in breast cancer can be determined using a mammosphere assay, the cells from each fraction were cultured as mammospheres under non-adherent conditions. The mammosphere assay revealed that the most enriched CSC fraction was FII (which will later be used as a CSC fraction), and the smallest contained fraction was FVI (which will later be used as a bulk cell fraction) (Appendix A). We demonstrated that the CSCs’ enriched fraction had 1.6–2 times more mammosphere compared to the bulk cell fraction (Figure 1B–D). Moreover, an increased colony formation capacity was observed for the CSC fraction compared to the bulk cell fraction from MDA-MB-468 (Appendix A). However, the CD44 assay did not reveal any difference in the CD44 level for MDA-MB-231 cells (Appendix A). Figure 1 shows that both the CSCs and bulk cell fractions each represented around 8% of the total cell population.

### 2.2. Fractionation of Mammospheres by Size Using Density Gradient Centrifugation

Next, the density gradient method was adapted for mammosphere separation by size. Compared to the separation of adherent cultured breast cancer cells, we increased the time from 30 min to 90 min and made the gradient steeper for mammosphere separation. After density gradient centrifugation, four fractions (FI-IV) were gently collected from the top to the bottom based on the density gradient (Figure 2A). The mammosphere separation efficiency was analyzed by fluorescence microscopy (Figure 2B–D). The microscope images show that before separation, the mammospheres were more heterogeneous in size (Figure 2B), whereas after separation, the size of the mammospheres in each fraction became more homogenous (Figure 2C,D). As expected, the count and size of mammospheres decreased by increasing the OptiPrep density (Figure 2, Appendix A). We found that FII contained the largest mammospheres and FV contained mammospheres of the smallest size (Figure 2E,F, Appendix A). The figure shows that in the case of Luminal A, the dimension of mammospheres decreased by 36% (Figure 2E), and in the case of TNBC, the mammosphere decreased 4.6 times in size (Figure 2F). Next, a secondary mammosphere assay was performed for the MDA-MB-231 cell line. CSCs from large mammospheres displayed more stemness-like properties, producing 46% more mammospheres compared to the CSCs from small mammospheres (Figure 2G). 

### 2.3. CSCs from TNBC Have Hybrid Metabolism and CSCs from Large Mammospheres Have Increased Glycolytic Capacity 

A recent study demonstrated that breast CSCs are hyper-metabolically active with an increased reserve capacity to survive under non-adherent growth conditions [7]. Therefore, the ATP level was measured in the CSCs and bulk cells. We did not find any difference in the ATP level between the CSCs and bulk cancer cells (Figure 3).

Next, the CSCs’ metabolic activity was analyzed more deeply. The metabolic flux analysis showed that mitochondrial basal respiration and ATP production were not altered in the CSCs compared to the bulk cells (Figure 4A,C,E). For both Luminal A and TNBC, increased mitochondria maximal respiration and mitochondrial reserve capacity were observed in the CSCs compared to the bulk cells (Figure 4A,C,E). The maximal respiration and reserve capacity increased by 16% and 32%, respectively in the Luminal A CSCs (Figure 4A). Similarly, the maximal respiration and reserve capacity increased by 13–15% and 11–29%, respectively, in the CSCs of TNBC (Figure 4C,E). However, an increased mitochondrial proton leak (25–30%) was only observed in the CSCs from TNBC (Figure 4C,E). Moreover, a glycolytic flux analysis was performed for the CSCs and bulk cell fractions (Figure 4B,D,F). The glycolytic flux analysis did not reveal any difference in the glycolytic activity between the Lumina A CSCs and bulk cells (Figure 4B). Our study demonstrates that the metabolic plasticity of TNBC CSCs was associated with increased OXPHOS and glycolytic activity (Figure 4C–F). The glycolytic flux analysis showed that the glycolytic activity of CSCs, isolated from MDA-MB-468, increased by 27%, whereas the glycolytic reserve capacity decreased compared to the bulk cells (Figure 4D). Furthermore, increased glycolytic activity and glycolytic reserve capacity were observed in the CSCs isolated from MDA-MB-231 compared to the bulk cells (Figure 4F). A hybrid metabolism is probably needed for CSCs to survive and grow under non-adherent growth conditions. 

Next, a metabolic analysis was performed for CSCs isolated from large and small mammospheres (Figure 4G,H). The metabolic flux analysis revealed that the mitochondrial proton leak was altered in CSCs from mammospheres. Namely, the mitochondrial proton leak was 40% higher in the CSCs isolated from large mammospheres compared to the cells isolated from small mammospheres (Figure 4G). Moreover, a glycolytic flux analysis was performed for CSCs from large and small mammospheres. The glycolytic flux analysis showed that the glycolytic activity and glycolytic reserve capacity were higher by 29% and 45%, respectively, in the CSCs from large mammospheres compared to the CSCs from small mammospheres (Figure 4H). The hypoxic environment of large mammospheres (based on previous study [23]) might represent the main cause why CSCs shift their metabolism toward glycolysis. 

### 2.4. HYOU1 Is Highly Expressed in Hyoxic TNBC and HYOU1 Is Related with Aggressiveness, Stemness, Hyper-Metabolic Activity, Metastasis and Poor Prognosis of Breast Caner

In the current study, we demonstrate that large mammospheres mimicking the tumor hypoxic environment (based on previous study [23]) contain CSCs with more stemness-like properties and increased glycolytic activity compared to CSCs from small mammospheres (Figure 2G and Figure 4H). To test the hypothesis that hypoxia is related with the aggressiveness of TNBC, a bioinformatic tool was used. For this purpose, a meta-analysis of *HYOU1* was performed by using clinical data on patients with breast cancer (Figure 5A–D). Firstly, the meta-analysis showed that TNBC is the most hypoxic breast cancer subtype (Figure 5B). Secondly, the meta-analysis revealed that *HYOU1* is overexpressed in TNBC, whereas the lowest *HYOU1* expression level is in Luminal A (Figure 5A). Altogether, an increased *HYOU1* expression level is correlated with the hypoxia level in TNBC (Figure 5A,B). Next, a meta-analysis was performed to identify how the alteration in the HYOU1 expression level affects the aggressiveness of breast cancer. The Kaplan–Meier analysis showed that an increased *HYOU1* expression level is associated with a decreased distant metastasis survival rate in lymph node-positive breast cancer (Figure 5C). Moreover, the meta-analysis demonstrated that among patients with breast cancer, *HYOU1* is overexpressed predominantly at the late stage of the tumor (Figure 5D). Next, we tested how the overexpression of *HYOU1* affects the stemness and metabolism of breast cancer cells. A mammosphere assay indicated that the overexpression of *HYOU1* increased the capacity to form a mammosphere by 60% in a Luminal A model (Figure 5E). We demonstrated that *HYOU1* overexpression not only increases stemness but also increases the metabolic activity of breast cancer cells. Indeed, a 23% higher ATP level was observed in overexpressed *HYOU1* cells compared to control MCF7 cells (Figure 5F).

## 3. Discussion

Based on the CSC theory, traditional chemotherapy mainly targets bulk cancer cells, while CSCs remain untouched, and as a result, CSCs may cause tumor relapse [24]. Several studies have shown that TNBC is the most CSC-enriched breast cancer subtype and an increased number of CSCs is correlated with poor prognosis [25]. Therefore, the ability to identify and isolate CSCs from the tumor will help to develop a new therapeutic strategy for aggressive TNBC. Currently, there are four methods available for the isolation and identification of CSCs from breast cancer [26]. The most used method to enrich CSCs is based on cell surface biomarkers by FACS or magnetic sorting. The cell surface markers CD44+/CD24- are widely used to identify and isolate CSCs from breast cancer, whereas CD44+/CD24-enriched CSCs have been found in mesenchymal TNBC [27]. Another CSC isolation method is a side population assay based on CSCs’ intrinsic properties [26]. The Hoechst assay is widely used to identify the CSC population. The main limitation of the Hoechst assay is the frequent false positive detection of cancer stem cells [20]. Moreover, aldehyde dehydrogenase assays are widely used to identify CSCs in different cancers, including breast cancer. The aldehyde dehydrogenase assay has revealed that ALDH1+ positive CSCs are predominantly located in epithelial TNBC [26]. Another feature of CSCs is the ability to grow under non-adherent conditions, forming spheroids. There is a large body of evidence that CD44+/CD24- and ALDH1+-enriched cells can form mammospheres [28,29,30]. The advantage of this assay is the potential to identify a novel CSCs population with a new biomarker from heterogenous spheroids [26]. The main disadvantage of the spheroid assay is its ability to identify only proliferative CSCs but not quiescent CSCs [31]. Therefore, there is a lack of any universal biomarkers or efficient methods to identify and isolate CSCs from TNBC. In the current study, we demonstrated that the density gradient centrifugation method is a simple and non-marker-based method to identify and isolate CSCs from TNBC and Luminal A. Recent studies demonstrated that density gradient centrifugation allows for the isolation of both quiescent and fast-cycling CSCs from the tumor [20,21]. A study on human glioblastoma showed that high-density glioblastoma cells are enriched with quiescence CSCs and treatment-resistance cancer cells [21]. Oppositely, our study demonstrates that the low-density fractions of TNBC and Luminal A are enriched with CSCs. Similarly, Liu et al. indicated that low-density fractions of rat hepatic tumor cells are CSCs-enriched and contain highly proliferative cells that are resistant to chemotherapy [20]. This indicates that the density gradient centrifugation method can be used for breast CSC isolation and for developing a novel drug treatment strategy for TNBC.

Accumulating evidence has indicated that CSCs can switch their metabolism between glycolysis and OXPHOS to maintain stemness and survive within a vulnerable tumor environment [25,32]. Recently, a hybrid metabolic model has been proposed for TNBC [8]. The current study shows that CSCs isolated from TNBC cells by density gradient centrifugation have increased glycolytic and OXPHOS activity. Interestingly, a similar metabolically hyper-active sub-population of CSCs has been isolated from TNBC by using FACS [33]. Namely, breast CSCs isolated by FACS have increased mitochondrial maximal capacity and mitochondrial leak, and CSCs consume more glucose compared to bulk cells. Our previous studies showed that TNBC has the highest mitochondrial activity among breast cancer subtypes, whereas a high mitochondrial mass in Luminal A is associated with poor prognosis [34,35]. The current study shows that both Luminal A and TNBC CSCs have increased mitochondrial maximal capacity and mitochondrial reserve capacity. It was recently proposed that spare respiration capacity is one of the key parameters of the aggressiveness of cancer cells [36]. Firstly, in vitro studies on a large amount of cell lines, including breast cancer cell lines, demonstrated that a high mitochondrial reserve capacity is a reason why cancer cells are chemotherapy-resistant [37]. Secondly, there is evidence that a high spare respiration capacity level helps cancer cells adapt to stress conditions more quickly. For example, cancer cells with a low spare respiration capacity are more sensitive to glucose deprivation [38]. Thirdly, our study shows that CSCs from aggressive TNBC have both high glycolytic reserve capacity and spare reserve capacity, which allow cancer cells to switch between glycolysis and OXPHOS energy states. Moreover, we demonstrated that TNBC CSCs have increased mitochondrial proton leak. Increased mitochondrial proton leak is probably associated with the overexpression of mitochondrial uncoupler UCP2. A recent study demonstrated that UCP2 overexpression increases breast cancer cells’ stemness [39]. In vitro and in vivo studies showed that the overexpression of UCP2 promotes breast cancer tumorigenesis [40]. AMP-dependent kinase (AMPK) is the main cellular sensor which is able to modulate mitochondria’s physiology, including mitochondrial reserve capacity and mitochondrial proton leak [41,42]. A high mitochondrial reserve capacity and increased AMPK activity are reasons why breast cancer cells are resistant to metformin and doxorubicin [41,43]. Recent studies demonstrated that AMPK activation regulates the stemness of breast cancer cells and AMPK modulates metabolic plasticity in TNBC [8,43]. 

Accumulating evidence shows that increased aerobic glycolysis is associated with tumor aggressiveness and multi-drug resistance [44,45,46]. A study on a breast tumor demonstrated that CSCs have higher glycolytic activity compared to bulk cells [47]. The current work shows that TNBC CSCs isolated from 2D and 3D (large) mammospheres have increased glycolytic activity with high glycolytic reserve and glycolytic reserve capacity. A study on lymphoma cells demonstrated that an increased glycolytic reserve decreased cancer cells’ sensitivity to glycolytic inhibitors [48]. Numerous studies have indicated that glycolysis-related enzymes play a key role in the chemotherapy resistance formation in cancer cells [49]. It has been demonstrated that in breast cancer Hexokinase II (HK2), the rate-limiting enzyme in the glycolytic pathway promotes cancer progression [50]. In vitro studies showed that HK2 overexpression increases cancer cells’ resistance to chemotherapy and that HK2 downregulation improves the radiosensitivity of TNBC [50,51]. Moreover, another work demonstrated that HK2 also has an important role in the stemness of cancer cells, including in breast cancer [23,52,53]. A study on breast cancer showed that the expression of phosphofructokinase-P (PFK-P), the second rate-limiting enzyme in glycolysis, correlates with poor prognosis [54]. Moreover, it was found that the overexpression of PFK-P increases breast cancer cells’ stemness [33]. In hypoxic tumors, the main modulator of glycolysis is hypoxia-inducible factor-1 (HIF-1) [55]. For example, GLUT1 and LDH expression alteration induced by HIF-1 is associated with chemotherapy resistance and the metastasis of cancer cells [56,57]. We propose that targeting both glycolysis (HIF-1) and OXPHOS (AMPK) is necessary to decrease the metabolic plasticity of TNBC. This action would significantly improve the efficiency of chemotherapy against aggressive TNBC. 

Hypoxia is a common feature of solid tumors that facilitates the formation of a CSC population, metastasis, metabolic rearrangement and tumor progression [58,59,60]. Although human breast tissue is well oxygenized (O_2_ level around 8.5%), during tumor development, the oxygen level decreases by six times (O_2_ level around 1.5%) [58]. Our meta-analysis showed that TNBC is the most hypoxic breast cancer subtype. Moreover, we demonstrated that cells from large mammospheres mimicking the hypoxic environment (based on a previous study [23]) display more stemness-like properties compared to cells isolated from small mammospheres. A study on breast cancer indicated that hypoxia increases the expression of ALDH (stemness marker) and remodels the metabolism of cancer cells towards glycolysis [12,55]. Our study shows that in hypoxic environments, CSCs shift their metabolism towards glycolysis. Interestingly, recent work demonstrated that while chronic hypoxia increases the stemness of breast cancer cells, acute hypoxia decreases it [12]. It has been found that chronic hypoxia usually occurs at later stages of breast cancer, promoting the resistance of CSCs to chemotherapy [13]. In the current study, we used breast cancer cells with overexpressed *HYOU1* to mimic a chronic hypoxia environment. We found that *HYOU1* overexpression increases the stemness of Luminal A cells via the hyperactivation of their metabolic state, and as a result, an increased intracellular ATP level was observed. Our recent work showed that breast cancer cells have an increased ATP level, whereas a high intracellular ATP level fuels the metastatic activity of TNBC cells [7,61]. Our meta-analysis indicated that *HYOU1* is overexpressed in TNBC. In addition, *HYOU1* overexpression is associated with a poor prognosis for metastatic breast tumors. The overexpression of *HYOU1* has previously been found in the prostate, non-small cell lung cancer and ovarian cancer. The same works demonstrated that *HYOU1* expression is correlated with poor prognosis [14]. Moreover, in the current study, we indicate that *HYOU1* expression correlates with the breast cancer stage. Another study recently demonstrated that long noncoding RNA HYOU1-AS facilitated TNBC progression via HYOU1 overexpression [62]. Thus, targeting hypoxic CSCs via HYOU1 should be the most promising therapeutic strategy to eradicate CSCs in aggressive TNBC.

In conclusion, the current study demonstrates that density gradient centrifugation is a non-marker-based approach which allows for metabolic flexible (normoxic) CSCs and glycolytic (hypoxic) CSCs from aggressive TNBC to be enriched. We proposed that targeting OXPHOS and the glycolytic reserve capacity (normoxic) as well as HYOU1 (hypoxic) should be the most promising therapeutic strategy to eradicate CSCs in aggressive TNBC.

## 4. Materials and Methods

### 4.1. Experimental Model Cell Lines

Human breast cancer cell lines (MCF7, MDA-MB-231 and MDA-MB-468) were obtained commercially from the American Type Culture Collection (ATCC). All cell lines were grown in Dulbecco’s Modified Eagle Medium (DMEM; GIBCO, Grand Island, NY, USA) supplemented with 10% FBS, 1% Glutamax and 1% Penicillin-Streptomycin. All cell lines were grown at 37 °C in 5% CO_2_.

### 4.2. Gradient-Based Breast CSCs and Hypoxic Mammosphere Separation

The protocol was performed for 2D monolayer human breast cancer cell lines (Figure 6A) according to the gradient-based cell separation method [22]. Briefly, culture media DMEM without FBS was used to dilute 60% OptiPrep stock to 40%. Different levels of OptiPrep in DMEM media were then prepared as follows: 2 mL overlaid by 24%, 15% and 5% from bottom to top in a 15 m falcon. The upper part of the tube was filled with 3 mL cell suspension (3–6 × 10^6^ cells) in completed DMEM medium without OptiPrep. The density gradient centrifugation was performed at 800× *g* for 30 min at room temperature. After centrifugation, different density fractions (0–24% OptiPrep DMEM medium) were collected and transferred into a new tube. Then, fractions were diluted in PBS to 15 mL and pelleted by centrifugation at 300× *g* for 5 min at room temperature.

The density gradient centrifugation protocol was also adapted for 3D-mammsophere separation (Figure 6B). Culture media DMEM/F-12 was used to dilute 60% OptiPrep stock to 40%. Different levels of OptiPrep in DMEM/F-12 media were then prepared as follows: 2 mL was overlaid by 40%, 5% and 2% from bottom to top in a 50 mL falcon. After 5 days, growing mammospheres were collected from T225 Flask by gentle centrifugation (200× *g*) for 5 min and resuspended in 3 mL in completed DMEM/F-12 medium without OptiPrep. Then, the mammosphere solution (3 mL) was placed on the upper part of the OptiPrep gradient. Density gradient centrifugation was performed at 800× *g* for 90 min at room temperature. After centrifugation, different density fractions (0–40% Optiprep DMEM/F-12 medium) were collected and transferred into a new tube. Then, fractions were diluted in PBS to 15 mL and pelleted by centrifugation at 300× *g* for 5 min at room temperature. Mammospheres’ separation efficiency was analyzed by fluorescence microscopy.

### 4.3. Three-Dimensional Mammosphere Growth and Three-Dimensional Mammosphere Formation Assay and Colony Formation Assay

After gradient-based separation, a single cell suspension was prepared using manual disaggregation (25-gauge needle) enzymatic disaggregation (1× Trypsin-EDTA, Sigma Aldrich, Burlington, MA, USA, cat. #T3924) for 10 min at 37 °C and manual disaggregation (25-gauge needle); 500/cm^2^ cells were plated in mammosphere medium (DMEM-F12/B27/20 ng/mL EGF/PenStrep antibiotics) under non-adherent conditions in six-well plates coated with 2-hydroxyethylmethacrylate (poly-HEMA, Sigma, Burlington, MA, USA, cat. #P3932). Cells were grown for 5 days and maintained in a humidified incubator at 37 °C at an atmospheric pressure in 5% (*v*/*v*) carbon dioxide/air. After 5 days of culture, 3D-mammospheres > 50 μm were counted using an eyepiece (“graticule”), and the percentage of cells plated which formed spheres was calculated and is referred to as percent mammosphere formation, and the value was normalized to one bulk cancer cell or small mammosphere (1 = 100% mammosphere formation efficiency).

Colony formation assay was performed following nature protocol [63]. After density gradient centrifugation, 200 cells were seeded in 6-well plates at 37 °C with 5% CO_2_ for 14 days. After 14 days, cells were stained with 0.5% crystal violet for 30 min at room temperature. Stained colonies were counted using open-source software ImageJ downloads from https://imagej.net/ij/download.html (accessed on 12 December 2023).

### 4.4. Fluorescent Microscopy Analysis

After mammosphere separation with density gradient centrifugation, a microscopy analysis was performed by analyzing mammospheres in live cell imaging with the EVOS imaging platform (ThermoFisher, Waltham, MA, USA). Mammospheres were labeled by nuclear dye Hoechst 33342, and fluorescence images were analyzed by using open-source software CellProfiler download via the website https://cellprofiler.org/previous-releases (accessed on 12 December 2023) (50 µm threshold for mammosphere size was used).

### 4.5. ATP Assay Using Cell-Titer-Glo

Cell-Titer-Glo (#G7570) was obtained from Promega, Inc. (Madison, WI, USA) and was used according to the manufacturer’s recommendations to measure intracellular ATP levels in lysed cells. Cell-Titer-Glo is a luciferase-based assay system. Luminescence content was evaluated using the Varioskan™ LUX plate reader (Thermo Scientific, Waltham, MA, USA) where the ATP level was normalized to nuclear fluorescent dye Hoechst 33342.

### 4.6. Metabolic Flux Analysis for Density Gradient Separated Breast Cancer Cells

Real-time oxygen consumption rates (OCRs) and extracellular acidification rates (ECARs) were determined using the Seahorse Extracellular Flux (XFe96) analyzer (Seahorse Bioscience, North Billerica, MA, USA) [64]. Briefly, after breast CSC separation by density gradient centrifugation, cells were washed with OCR or ECAR media, and single cells were counted in the presence of Trypan blue to assay cell vitality. After 3D mammosphere separation by density gradient centrifugation, mammospheres were dissociated into single cells by trypsin (incubation for 10 min at 37 °C) and passed through a syringe. After cells were washed with OCR or ECAR media, they were counted in the presence of Trypan blue. Then, density gradient separated single cells (20–30 × 10^3^) were transferred into a Cell-Tak pre-coated 96-well XF microplate. The XF plate was centrifugated at 200× *g* for 1 min to accelerate the cells’ attachment to the bottom of the wells. Finally, the OCR and ECAR were measured using the Seahorse Metabolic Flux Analyzer (XFe96, North Billerica, MA, USA), under standard conditions at 37 °C, and at the end, both measurements were normalized for cell content by a fluorescent assay of the nuclear dye Hoechst 33342.

### 4.7. Viral Transduction of HYOU1 Gene

Lentiviral construct (GeneCopoeia, Rockville, MD, USA), ORF expression clone for HYOU1 ((NM_001130991.1) #EX-S0360-Lv10) was amplified, and LentiSuite™ Basic Kit (System Bioscences, Palo Alto, CA, USA, #LV340A-1) was used to create the stably transduced HYOU1 cell line from MCF7 cells following the manufacturer’s protocol.

### 4.8. Bioinformatics Analysis

HYOU1 gene analysis of primary breast cancer was based on the Cancer Genome Atlas (TCGA) platform (http://www.cbioportal.org/ (accessed on 12 December 2023)), where the METABRIC and Breast Invasive Carcinoma PanCancer Atlas databases were used.

To perform a Kaplan–Meier (K–M) analysis on the HYOU1 gene, we used an open access online survival analysis tool [65]. We primarily analyzed data from patients with lymph node-positive breast cancer. The hazard ratios of patients with breast cancer were calculated using the best auto-selected cutoff. K–M curves were created using the online K–M plotter (https://kmplot.com/analysis/index.php?p=service&cancer=breast (accessed on 12 December 2023)). The latest 2023 version of the database was utilized for all of these analyses.

### 4.9. Statistical Analysis

Data are expressed as mean ± SEM of over ≥3 independent experiments with ≥3 technical replicates per experiment. Differences between experimental groups were determined by the paired two-tailed Student’s *t*-test, and for K-M analysis, Log-rank test was performed. A *p*-value of less than 0.05 was considered statistically significant.

## Figures and Tables

**Figure 1 ijms-25-08958-f001:**
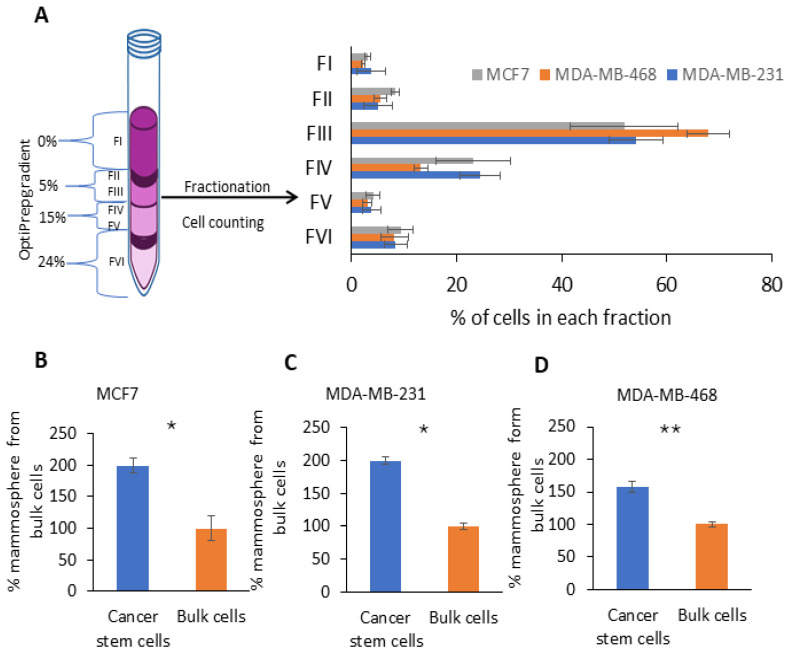
Breast cancer cell separation and mammosphere assay. (**A**) Schematic illustration of Optiprep gradient used for breast cancer cell fractionation and percentages of cells segregated in each fraction. Anchorage-independent growth quantification for fractions FII (cancer stem cells) and FVI (bulk cells) in (**B**) MCF7, (**C**) MDA-MB-231 and (**D**) MDA-MB-468 using 3D mammosphere formation assay. Bars are presented as mean ± SEM, and statistical analysis is carried out with paired two-tailed Student’s *t*-test; *n* = 3–4, and * and ** indicate statistically significant differences between the mean values, denoted as *p* < 0.05 and *p* < 0.01, respectively.

**Figure 2 ijms-25-08958-f002:**
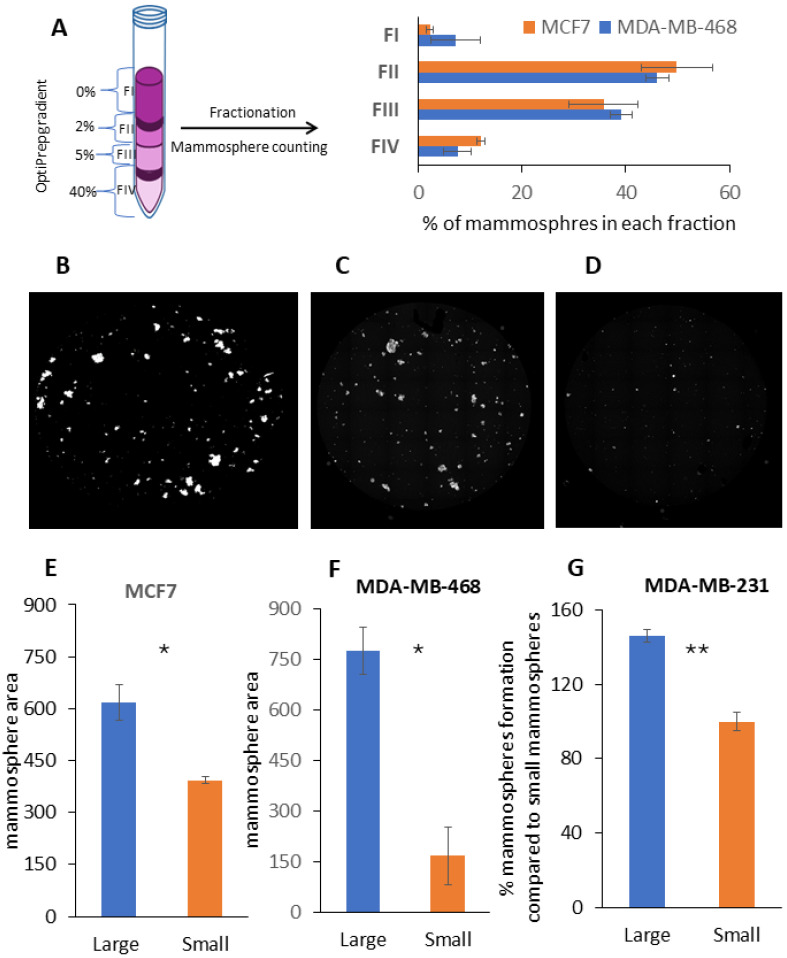
Mammosphere separation, mammosphere size distribution within density gradient and mammosphere assay. (**A**) Schematic illustration of Optiprep gradient used for mammosphere fractionation and percentages of mammospheres segregated in each fraction. Illustrative fluorescence microscopy figures of mammosphere distribution before mammosphere separation (**B**) and after mammosphere separation fractions (**C**) FII and (**D**) FIV. Mammosphere size (mammosphere area pixels) assay for (**E**) MCF7 and (**F**) MDA-MB-468 with fractions FII (large) and FIV (small) using fluorescence microscopy at 40X magnification and CellProfiler software. Anchorage-independent growth quantification for fractions FII (Large) and FIV (Small) in (**G**) MDA-MB-231 using 3D mammosphere formation assay. Bars represent mean ± SEM. Statistical analysis was carried out using paired two-tailed Student’s *t*-test; *n* = 3–5. * and ** indicate statistically significant differences between mean values, denoted as *p* < 0.05 and *p* < 0.01, respectively.

**Figure 3 ijms-25-08958-f003:**
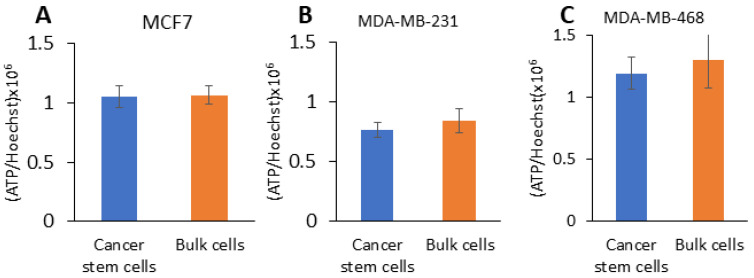
ATP level assay in fractionated breast cancer cells. ATP level was determined in (**A**) MCF7, (**B**) MDA-MB-231 and (**C**) MDA-MB-468 cancer stem cells and bulk cells using Cell-Titer-Glo. Bars represent mean ± SEM, and *n* = 3.

**Figure 4 ijms-25-08958-f004:**
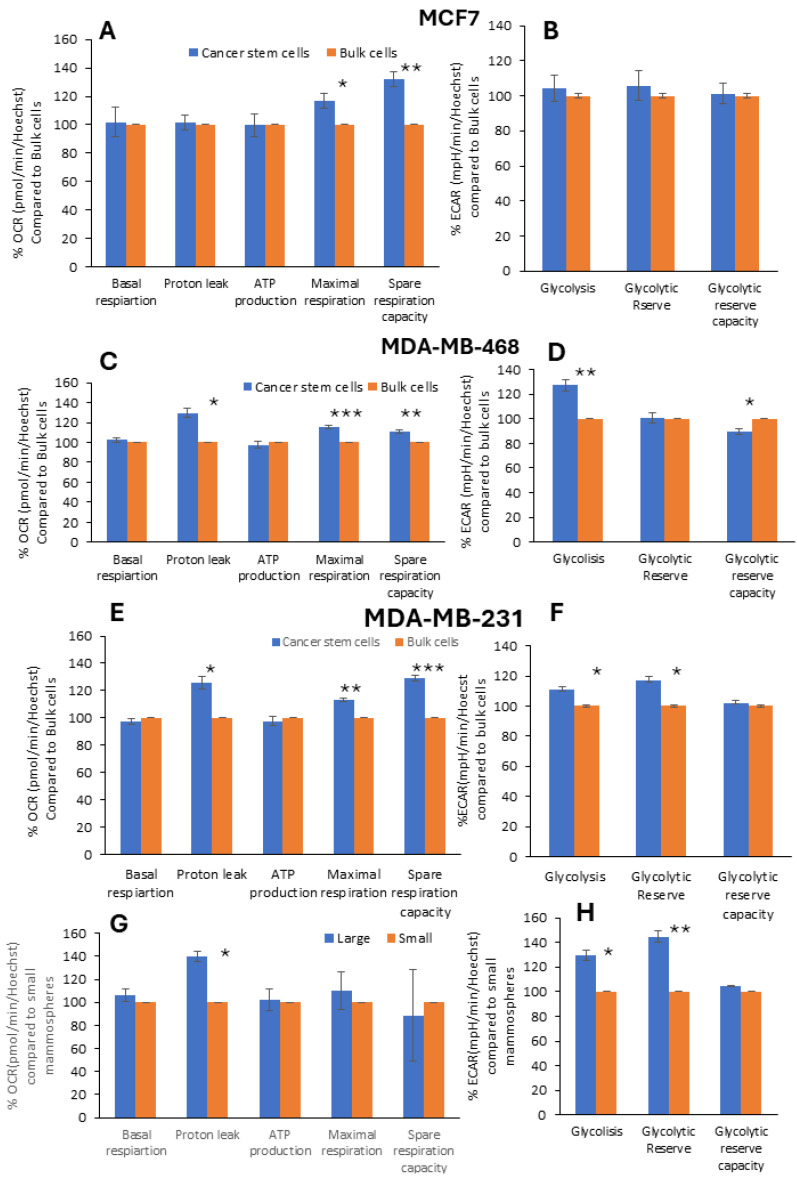
Metabolic flux analysis of breast cancer stem cells and mammospheres separated by density gradient centrifugation. OCR (oxygen consumption rate) and ECAR (extracellular acidification rate) were measured using Seahorse XFe96. Comparative analysis of (**A**) mitochondrial respiration (OCR) and (**B**) glycolysis (ECAR) with cancer stem cells and bulk cells isolated from MCF7 cells. Comparative analysis of (**C**) mitochondrial respiration (OCR) and (**D**) glycolysis (ECAR) with cancer stem cells and bulk cells isolated from MDA-MB-468 cells. Comparative analysis of (**E**) mitochondrial respiration (OCR) and (**F**) glycolysis (ECAR) with cancer stem cells and bulk cells isolated from MDA-MB-231 cells. Comparative analysis of (**G**) mitochondrial respiration (OCR) and (**H**) glycolysis (ECAR) with large and small mammospheres from MDA-MB-231 cells. Bars represent mean ± SEM, and paired two-tailed Student’s *t*-test was used for statistical analysis; *n* = 3–4. *, ** and *** indicate statistically significant differences between mean values, denoted as *p* < 0.05, *p* < 0.01 and *p* < 0.001, respectively.

**Figure 5 ijms-25-08958-f005:**
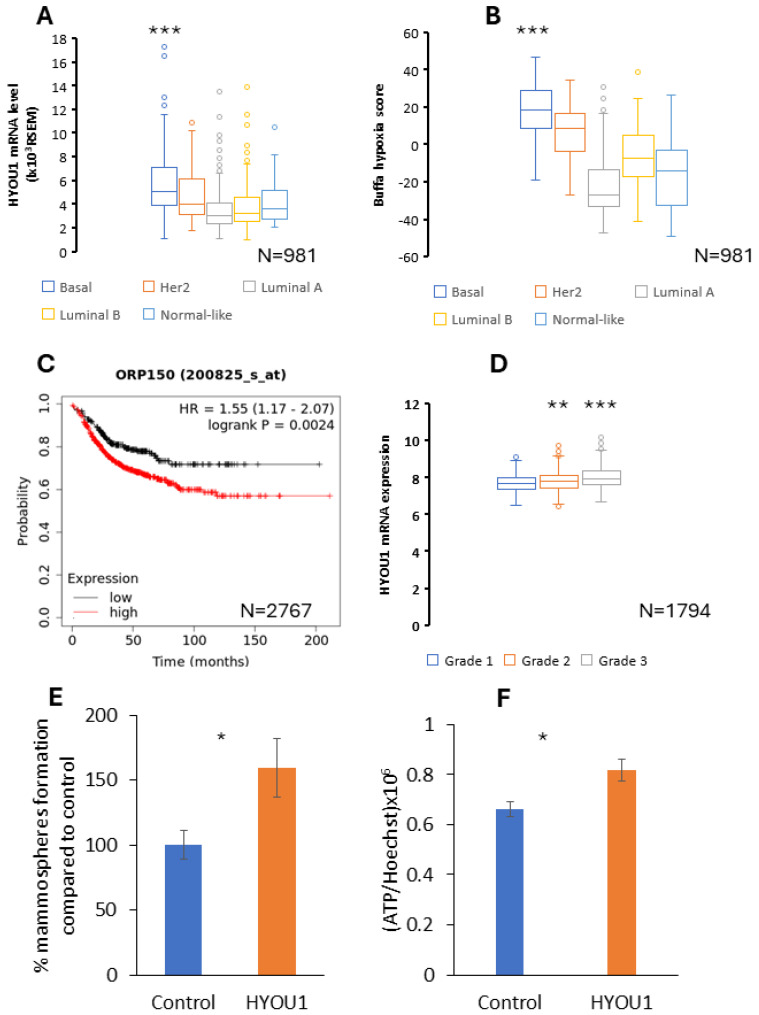
HYOU1 is highly expressed in triple-negative breast cancer (basal) and hypoxic breast cancer, and HYOU1 expression is associated with the aggressiveness of breast cancer. The Cancer Genome Atlas database was utilized to analyze (**A**) the HYOU1 pattern, (**B**) Buffa hypoxia score and (**D**) how HYOU1 mRNA expression is associated with the tumor grade in breast cancer (a statistic analysis was carried out using the unpaired two-tailed Student’s *t*-test). A Kaplan–Meier survival analysis showing the relationship between (**C**) HYOU1 (ORP150) mRNA expression and distant metastasis-free survival (DMFS) in patients with lymph node-positive (LN(+)) breast cancer (a statistical analysis was performed using the Log-rank test). (**E**) A mammosphere assay was carried out for HYOU1-transfected MCF7 cells and (**F**) an ATP assay was carried out for HYOU1-transfected MCF7 cells using Cell-Titer-Glo (the bars present the mean ± SEM, and a statistic analysis was carried out using the paired two-tailed Student’s *t*-test *n* = 3). *, ** and *** indicate statistically significant differences between the mean values, denoted as *p* < 0.05, *p* < 0.01 and *p* < 0.001, respectively.

**Figure 6 ijms-25-08958-f006:**
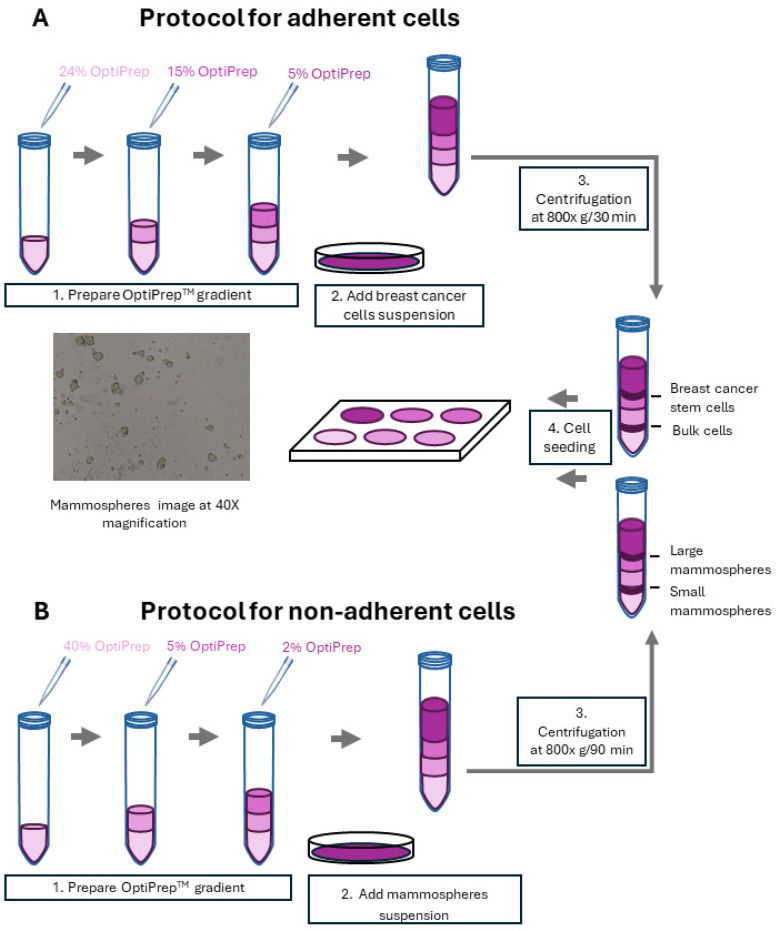
Density gradient centrifugation protocols for adherent (**A**) and non-adherent (**B**) breast cancer cells to isolate cancer stem cells. After gradient-based separation, single cell suspension was prepared using manual disaggregation (25-gauge needle) or after mammosphere gradient-based separation enzymatic assay (1× Trypsin-EDTA) for 10 min at 37 °C, and manual disaggregation was carried out (25-gauge needle); 500/cm^2^ cells were plated in mammosphere medium.

## Data Availability

Request for raw data can be directed to the corresponding authors.

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
