# Peer review of "Density Gradient Centrifugation Is an Effective Tool to Isolate Cancer Stem-like Cells from Hypoxic and Normoxia Triple-Negative Breast Cancer Models"

_ijms, 2024, doi:10.3390/ijms25168958_

Round 1

Reviewer 1 Report

Comments and Suggestions for Authors

Comments

This study proposes density gradient centrifugation as a method for the isolation of CSCs from cancer breast models. The authors obtained enriched fractions of CSCs for Luminal and TNBC cells evaluated for the sphere formation. In these enriched fractions, changes in energy metabolism were observed. In addition, with a bioinformatics analysis, the relation of HYOU1 with the aggressiveness, metastasis, and poor prognosis of TNBC was analyzed. In vitro, the overexpression of HYOU1 increases the stemness and modifies the metabolism in breast cancer cells. However, additional experimental evidence is necessary to support the conclusions.

Major changes

P.3, lines 110-112. It is recommended that the proportion of the mammospheres in each fraction be shown to understand why fraction II is the most enriched CSC fraction. Why was the bulk cell fraction fraction FVI and not the rest of all fractions? In addition, to validate the enrichment of CSCs, it is necessary to determine the levels of CD44+/CD24-markers and make additional assays such as colony-forming ability, drug resistance, and tumorigenicity in vivo.

Figure 4. What is the physiological advantage of increased maximal and spare respiration in CSCs?

P. 4, lines 134-135. Also, it is recommended that the size of mammospheres for each fraction be shown.

P.7, lines 188-189. What experimental evidence indicates a hypoxic environment exists in large mammospheres, not small ones? What are the levels of HIF1α in large and small mammospheres?

P.9, section 2.4. The meta-analysis suggested high levels of HYOU1 in TNBC but low levels in Luminal A. However, the HYOU1 levels should be determined in the MDA-MB-468, MDA-MB-231, and MCF7 to validate this prediction. In addition, the relation of HYOU1 levels with the enrichment of CSCs, CSC metabolism, and hypoxia was not determined. All these experiments are indispensable to understanding the relation of HYOU1 with hypoxia, stemness, and metabolism.

P.11, lines 261-262. This study did not validate that cancer cells obtained by density gradient are effective CSCs. Other assays were not realized, such as CD44+/CD24 markers, colony-forming ability, drug resistance, and tumorigenicity in vivo.

P.12, lines 326-327. It was not demonstrated that there is a hypoxic environment in the mammospheres.

Minor changes

P.3, lines 107-108. Fraction III contained more proportion of cells than fraction II.

Figure 2E, 2F and 2G. Which are the units of the mammospheres area? Figure 3. Which are the units of ATP?

Methods. Describe more details about the adaptation of density gradient centrifugation protocol to breast cancer models.

P16, line 383-386. When was the cellular suspension obtained for manual disaggregation or separation enzymatic? What number of cells were used in the formation of mammospheres?

Correct typographical mistakes

Author Response

Thank you for your comments and suggestions which helped to improve current manuscript.

Major changes

Comments 1. P.3, lines 110-112. It is recommended that the proportion of the mammospheres in each fraction be shown to understand why fraction II is the most enriched CSC fraction. Why was the bulk cell fraction fraction FVI and not the rest of all fractions? In addition, to validate the enrichment of CSCs, it is necessary to determine the levels of CD44+/CD24-markers and make additional assays such as colony-forming ability, drug resistance, and tumorigenicity in vivo.

Answer: Figure S1-S3 all fractions mammosphere numbers were added and   CD44+ assay was added (Figure S4).

Comment 2. Figure 4. What is the physiological advantage of increased maximal and spare respiration in CSCs?

Answer: It was recently proposed that spare respiration is one of the key parameters of aggressiveness of cancer cells. Firstly, it was discovered that increased mitochondrial reserve capacity correlates with metformin resistance formation in cancer cells. Secondly, there is evidence that high spare respiration capacity level helps faster adapt cancer cells in stress condition. For example, cancer cells with low spare respiration capacity are more sensitive to glucose deprivation. Thirdly, our current study showed that CSCs from aggressive TNBC have both high glycolytic reserve capacity and spare reserve capacity which allow cancer cells switch between energy states glycolysis and OXPHOS (Line 308-316)

Comment 3. 4, lines 134-135. Also, it is recommended that the size of mammospheres for each fraction be shown.

Answer: For imaging analysis, we added threshold 50 µm. It means that small fraction contain mammaspheres < 50 µm and large >50 µm. We did not directly measured size of mammaospheres. According to the literature MCF7 cells mammosphere size can reach to 300 µm at day 5. We added mammosphere area for each fraction (Figure S5 and Figure S6)

Comment 4. P.7, lines 188-189. What experimental evidence indicates a hypoxic environment exists in large mammospheres, not small ones? What are the levels of HIF1α in large and small mammospheres?

Answer: In current manuscript we have not measured HIF1α level in mammosphere. According to the literature, spheroids of 300–500 µm of size are those that best mimic in vivo tumors in terms of hypoxia. It was demonstrated that core of mammosphere is hypoxic area where increased HIF1α was noticed. Hypoxic enviroment is based on previous study breast mammosphere (reference was added) Line 208).

Comment 5. P.9, section 2.4. The meta-analysis suggested high levels of HYOU1 in TNBC but low levels in Luminal A. However, the HYOU1 levels should be determined in the MDA-MB-468, MDA-MB-231, and MCF7 to validate this prediction. In addition, the relation of HYOU1 levels with the enrichment of CSCs, CSC metabolism, and hypoxia was not determined. All these experiments are indispensable to understanding the relation of HYOU1 with hypoxia, stemness, and metabolism.

Answer: It was found that during acute hypoxia (6h) increased HIF1α in CSCs of breast cancer cells whereas during chronic hypoxia shifts towards HYOU1 overexpression (Ernestina Marianna De Francesco et al.,Oncotarget, 2017, Vol. 8, (No. 34)) . Same work demonstrated that chronic hypoxia increase breast cancer stemness ( CSC marker ALDH) which is related with increased mitochondrial mass. In the next manuscript we are planning to study how HIF1α (acute hypoxia) and HYOU1 (chronic hypoxia) affect metabolic plasticity of breast CSCs how it is related with aggressiveness, drug resistance and stemness of breast cancer cells by using Density gradient centrifugation method.

Comment 6. P.11, lines 261-262. This study did not validate that cancer cells obtained by density gradient are effective CSCs. Other assays were not realized, such as CD44+/CD24 markers, colony-forming ability, drug resistance, and tumorigenicity in vivo.

Answer: CD44 assay has done and CD44 assay data is added in supplement (Figure S4). Several works demonstrated that density gradient centrifugation method is an efficient tool to isolate CSCs from primary hepatic cancer rat model  and human primary glioblastoma tumor (Reference 20,21). Study on hepatic rat cancer model density gradient centrifugation method allow to isolate CSCs with increase stemness markers EpCAM and CD133, there are drug resistance and this population of cells are tumorigenic in vivo. (Reference 20) In case of human primary glioblastoma tumor CSCs isolated by density gradient centrifugation method are drug treatment resistance (Reference 21).

Comment 7. P.12, lines 326-327. It was not demonstrated that there is a hypoxic environment in the mammospheres.

Answer: This is based on previous study. Reference was added. Line 359

Minor changes

Comment 1. P.3, lines 107-108. Fraction III contained more proportion of cells than fraction II.

Answer: Yes, Fraction III contained more cells than fraction II. We corrected this in text. line 116

Comment 2. Figure 2E, 2F and 2G. Which are the units of the mammospheres area? Figure 3. Which are the units of ATP?

Answer: Mammosphere area unite is pixels in figure 2 (E and F). ATP unite is Luminescence signal normalized to nuclear fluorescent signal.

Comment 3. Methods. Describe more details about the adaptation of density gradient centrifugation protocol to breast cancer models.

Answer:  Current density gradient centrifugation protocol has already detailed described in several studies (Reference 20-22).

Comment 4. P16, line 383-386. When was the cellular suspension obtained for manual disaggregation or separation enzymatic? What number of cells were used in the formation of mammospheres?

Answer: After gradient-based separation a single cell suspension was prepared using manual disaggrega-tion (25-gauge needle) or after mammosphere gradient-based sepa-ration enzymatic (1x Trypsin-EDTA) for 10 minutes at 37°C, and manual disaggregation (25-gauge needle). 500/cm2 cells were plated with in mammosphere medium (see line 405-408).

Comment 5. Correct typographical mistakes.

Answer: Typograpic mistakes were corrected.

Reviewer 2 Report

Comments and Suggestions for Authors

The article is very interesting and written in an appropriate way. The data and all the analyses are presented appropriately. The results are significant, and the overall work has potential applications.

Minor comments/corrections:

If possible, place the units of the mammosphere area in figure 2 (E and F).

Additionally, add a few sentences at the end of the discussion about how the current work can be translated or applied to real patient samples and clinical practice.

End of comments.

Author Response

thank you for your comments and suggestion which helped to improve current manuscript

Minor comments/corrections:

Comment 1. If possible, place the units of the mammosphere area in figure 2 (E and F).

Answer: Mammosphere area unite is pixels in figure 2 (E and F).

Comment 2. Additionally, add a few sentences at the end of the discussion about how the current work can be translated or applied to real patient samples and clinical practice.

Answer: In conclusion, current study demonstrated that density gradient centrifugation is a non-marker-based approach which allows to enrich metabolic flexible (normoxic) CSCs and glycolytic (hypoxic) CSCs from aggressive TNBC. We proposed that targeting OXPHOS and glycolytic reserve capacity  (normoxic) as well as HYOU1 (hypoxic) should be the most promising therapeutic strategy to eradicate CSCs in aggressive TNBC.

Round 2

Reviewer 1 Report

Comments and Suggestions for Authors

This reviewer considers that some queries could have been answered satisfactorily.

Although the authors showed the proportion of the mammospheres in each fraction, the statistical analysis was not made in supplementary figures to support that fraction II has a significant concentration of cancer stem cells. To validate the enrichment of CSCs, the authors only determined the CD44+ marker. However, it is necessary to show the combination of CD44+/CD24-markers and make additional assays such as colony-forming ability, drug resistance, and tumorigenicity in vivo. Nevertheless, the authors argue that these assays are not indispensable, citing two previous studies. These works are not comparable because the cancer cells used were different and taken from other sources. In addition, additional markers were determined in both works to validate their results, as suggested for this study with breast cancer cells.

Figures S5 and S6. The statistical analysis is indispensable to supporting the existence of significant cancer stem cells in fraction II.

Author Response

1) We agree with reviewer that cited are not comparable because the cancer cells used were different and taken from other sources. Thank you for suggestion which helps to improve our manuscript. We added colony formation assay (Figure S4, lines 123-124 and lines 496-502). We found that increased colony formation capacity is higher in CSCs fraction compared to bulk cells fraction from MDA-MB-468 (Figure S4). 

2) We performed statistic analysis for Figure S6 and Figure S7 data and showed that significant statistic difference is only between FII and FIV which confirm that FII contain the largest mammospheres.

Round 3

Reviewer 1 Report

Comments and Suggestions for Authors

This reviewer has the following suggestions: 

Include figures S1-S3 in the main text and add statistical analysis to these figures.

Figures S4 and S6-S7 should also be incorporated into the main text.

 Incorporate the colony formation assay in the methods section.

Author Response

Thank you for suggestions.

1) Figures S1-S3 in the main text have included (line 113) and statistics analysis were performed, as well

2) Figures S4 (line 116)  and S6-S7 (line 140) have incorporated in main text . 

3) Colony formation assay was incorporated in in the methods section.